# The Antimicrobial Applications of Nanoparticles in Veterinary Medicine: A Comprehensive Review

**DOI:** 10.3390/antibiotics12060958

**Published:** 2023-05-25

**Authors:** Mariana Paiva Rodrigues, Priscila Natália Pinto, Raul Roque de Souza Dias, Gabriela Lago Biscoto, Lauranne Alves Salvato, Ruben Dario Sinisterra Millán, Ricardo Mathias Orlando, Kelly Moura Keller

**Affiliations:** 1Programa de Pós-Graduação em Ciência Animal, Escola de Veterinária, Universidade Federal de Minas Gerais, Belo Horizonte 30270-901, Brazil; rodrigues.mpaiva@gmail.com (M.P.R.); priscilanatalia24@gmail.com (P.N.P.); diasrrs1@gmail.com (R.R.d.S.D.); 2Departamento de Medicina Veterinária Preventiva, Escola de Veterinária, Universidade Federal de Minas Gerais, Belo Horizonte 30123-970, Brazil; gabrielabiscoto@gmail.com (G.L.B.); lausalvato@gmail.com (L.A.S.); 3Departamento de Química, Instituto de Ciências Exatas, Universidade Federal de Minas Gerais, Belo Horizonte 30123-970, Brazil; sinisterra@ufmg.br (R.D.S.M.); orlandoricardo@hotmail.com (R.M.O.)

**Keywords:** antibiofilm, antimicrobial, antimycotoxin, drug delivery system, nanostructures

## Abstract

Nanoparticles (NPs) are nanoscaled particles sized from 1–100 nm, which can be composed of inorganic or organic compounds. NPs have distinctive morphology, size, structure, and surface features, which give them specific properties. These particular attributes make them interesting for biological and medical applications. Due to these characteristics, researchers are studying the possible aptness of numerous nanoparticles in veterinary medicine, such as the capacity to act as a drug delivery system. The use of these NPs as a possible bactericidal or bacteriostatic medication has been studied against different bacteria, especially multiresistant strains and the ones that cause mastitis disease. The antibiofilm property of these nanostructures has also already been proved. The antiviral activity has also been shown for some important viral animal diseases; the antifungal activity had been demonstrated against both pathogenic and mycotoxigenic species. Therefore, this review aimed to elucidate the main clinical and preventive veterinary applications of inorganic and organic nanoparticles.

## 1. Introduction

Nanoscience comprehends the studies, discoveries, and understanding of size- and structure-dependent properties, especially nanoscale subjects. The scientific knowledge to synthesize, control, and manipulate these nanoscale structures is called nanotechnology [1]. The study of this technology for biological and pharmaceutical applications has been conducted since the 1950s, and the first nanoparticle for vaccines and drug delivery purposes was developed in the late 1960s [2].

Nanoparticles (NPs), one of the main categories of nanoobjects, have all three external dimensions in the nanoscale, ranging from 1–100 nm. NPs are commonly classified by different parameters, composition, morphology, structure, and surface features. These characteristics influence magnetic, optical, electronic, biological, and environmental interactions and other properties [3]. The three main classes of NP are polymeric, lipid-based, and inorganic NPs and the size and high surface area–volume ratio characteristics are common to all three. However, each one has particular parameters which give them different attributes, advantages, and disadvantages [4]. Nanoparticles have been studied for many years for different applications and purposes. As will be shown in this paper, NPs have antibiotic, antifungal, antimycotoxigenic, antibiofilm, antiparasite, antiviral, and drug delivery system properties. These promising features evidence the numerous possibilities and future applications of NPs in the veterinary medicine field. To understand the impact of nanoparticles in the veterinary medicine field, we conducted a basic bibliometric evaluation. For that analysis, the keywords “nanoparticles AND veterinary” with a restriction filter for the year (from 2013 to 2023) were used. The document count over time during the period established and obtained using the Lens.org database can be observed in Figure 1. With this graphic it is possible to observe the increase in publications on nanoparticles in the veterinary field, which elucidates the high impact and importance of this comprehensive review.

Among all the documents found using the research terms “nanoparticles AND veterinary” it was possible to find the 50 countries with the highest number of publications from 2013 to 2023. These results are shown in the following word cloud (Figure 2).

Despite the high number of studies found using these keywords and database, the majority of the NP studies are basic research, focusing on microorganisms, independently of whether they are a major cause of human or animal diseases. Therefore, this comprehensive review aimed to comprise research papers that evaluated the antimicrobial effects of different NPs against species with a relevant impact in the clinical and preventive fields of veterinary medicine.

## 2. Relevant Sections

### 2.1. Antibacterial Activity

Bovine mastitis is a pathology of great clinical and economic relevance in veterinary medicine. This disease causes harm to the welfare of the affected animal, direct and indirect damage to milk production, in addition to reducing the quality and food safety of the product. Due to the increase in the emergence of multiresistant strains, bovine mastitis represents an even greater risk for the animals and producers [5]. Among the contagious mastitis pathogens, *Streptococcus agalactiae* and *Staphylococcus aureus* were considered, for a long time, the most important; however, in recent years a significant increase in environmental and opportunistic microorganisms causing mastitis has been detected [6].

Nanoparticles (NPs) have been considered an innovative and effective alternative for the treatment of mastitis caused by contagious or opportunistic bacteria and fungi, especially multidrug-resistant ones. Silver nanoparticles (Ag NPs) showed bactericidal and bacteriostatic characteristics against multiresistant strains of *S. agalactiae* and *S. aureus* in different studies [7,8,9]. Other opportunistic mastitis-causing bacteria and fungi were also sensitive to treatments with Ag NPs [7,9,10].

Strains of *S. agalactiae* were tested against Ag NPs, which were spherical in shape and ranged from 15 to 50 nm and had a 50% and 90% minimum inhibitory concentration (MIC50 and MIC90) of 8 and 16 μg mL^−1^, respectively; also, the combination of Ag NPs and cinnamon oil showed improved activity against the *S. agalactiae* strains [8]. The MIC of Ag NPs and silver–copper complex nanoparticles (Ag-Cu NPs) found was 3.125 μg mL^−1^ for *S. agalactiae*, 6.25 μg mL^−1^ for *S. dysagalactiae*, *Salmonella* spp., *Enterococcus faecalis*, *Enterobacter cloacae*, and *Escherichia coli*, and 12.5 μg mL^−1^ for *S. aureus*. These differences in action among the two tested nanoparticles are due to the sizes of both NPs. Cu NPs showed the biggest average size of 300 nm while, on the other hand, Ag NPs’ average size was 150 nm [9]. Agreeing with the previously cited paper, a reduction in the bacterial viability was obtained using Ag NPs against *Enterococcus faecalis*, *Enterobacter cloacae*, *S. agalactiae,* and *Escherichia coli*. On the other hand, this same paper showed that the average size of Ag NPs was smaller than the average size of Cu NPs, 262 and 288.6 nm, respectively. However, this paper also showed a higher agglomeration tendency of the Cu NPs [7]. Biosynthesized Ag NPs, derived from *Curcuma longa* (ClAg NPs) or *E. coli* (EcAg NPs), presented an antibacterial effect, with MICs of 71.8 and 0.438 μM for *P. aeruginosa* and 143.7 and 3.75 μM for *S. pseudintermedius*, respectively. In addition, the study showed a significant synergistic effect of these nanoparticles when in combination with carbenicillin used to treat *P. aeruginosa* and ampicillin used for *S. pseudintermedius* [11].

The efficiency of the antimicrobial activity of Ag NPs can be improved by combining them with other compounds, such as other nanoparticles and/or biological compounds. Silver nanoparticles decorated with quercetin nanoparticles (QA NPs) significantly reduced the viability of *E. coli* strains isolated from cows with mastitis in a dose-dependent concentration; the survival rate of cells treated with 10 μg mL^−1^ was 0% [12]. Another example of the combination of compounds and synergistic effects in the control of mastitis is the complex of silver and copper nanoparticles (Ag-Cu NPs) which was more effective than Ag NPs and copper nanoparticles (Cu NPs) alone [7,9].

Zinc oxide nanoparticles (ZnO NPs), as well as Ag NPs, show inhibitory effects against microorganisms that can cause mastitis. The antimicrobial effects of ZnO NPs and capped ZnO NPs were tested against *S. aureus*, *E. coli*, and *Klebsiella pneumonia*. Despite the bigger size of capped ZnO NPs, these nanoparticles showed an improved antibiotic capacity for *S. aureus* and *Klebsiella pneumonia,* pathogenic bacteria isolated from milk from cows with clinical mastitis. The MIC of capped and uncapped ZnO NPs was 20 μg mL^−1^ and no effect, <1 μg mL^−1^ and 10 μg mL^−1^, for *S. aureus* and *Klebsiella pneumoniae*, respectively. However, for *E. coli* both NPs showed the exact same MIC, 5 μg mL^−1^ [13]. Spherical ZnO NPs, of an average size of 30 nm, also showed desirable inhibitory and bactericidal effects against multidrug-resistant *S. aureus* and *E. coli* bacteria isolated from the milk of ewes affected by mastitis. The study also demonstrated a greater effect of ZnO NPs against Gram-positive (G+) than Gram-negative (G−) bacteria. The minimum bactericidal concentration (MBC) and MIC were 3.9 μg mL^−1^ and 7.81 μg mL^−1^ for *S. aureus* and 31.25 μg mL^−1^ and 62.5 μg mL^−1^ for *E. coli*, respectively [14]. Another application of these ZnO NPs, as well as zinc oxide quantum dots (ZnO QDs), is the use of these compounds conjugated to silica to facilitate the detection of clinical and subclinical mastitis through the amplification of fluorescence detection of inflammatory biomarkers of mastitis [15].

Other materials can also be used in the nanoparticulate or conjugated or encapsulated form, such as propolis encapsulated in nanoparticles. The effect against the growth of *S. aureus* and other bacteria isolated from the milk of animals with mastitis was variable according to the formulation [16]. Chitosan is a biological polysaccharide molecule derived from chitin that demonstrated antibacterial properties against bacteria mastitis isolates [17]. Due to this, Aguayo et al. (2020) studied the formulation of chitosan nanoparticles (Qo NPs) and the growth of *Pseudomonas* sp. isolated from milk from cows with clinical mastitis and found that the antimicrobial capacity of the formulation was equal to or greater than conventional antibiotics [18]. Furthermore, in another study, it was proven that smaller particles of Qo NPs have better antimicrobial activity on the development of *S. aureus* and *S. xylosus* strains [19].

In addition to the potential use of nanoparticles in the treatment of mastitis, their use to control and prevent other microorganisms, multiresistant or not, that cause veterinary pathologies has been proven. The MIC and MBC of spherical Ag NPs, with an average size of 10 nm, were 0.8 and 1.0 μg mL^−1^ for *Prevotella melaninogenica* and 1.0 and 1.5 μg mL^−1^ for *Arcanobacterium pyogenes*. This paper also showed that cell viability decreased in a dose-dependent manner [20]. This paper demonstrated the very low concentrations necessary to inhibit bacterial growth, and it is possible to infer that this could be correlated with the fact that the nanoparticles used had a small size, which can improve the antibacterial activity due to the enhancement in capacity to penetrate the bacterial membranes.

Different susceptibility to silver nanoparticles has been found between G− and G+ bacteria by some authors; this fact can be explained by the structural differences in the peptidoglycan layer of the cell wall. Cubic Ag NPs with an average size of 5 nm promoted a higher reduction in cell viability and lower MIC values for the G− *Pseudomonas aeruginosa* and *Shigella flexneri* in comparison to the G+ *Staphylococcus aureus* and *Streptococcus pneumoniae*. This same paper has also shown the positive combination of Ag NPs and six antibiotics (ampicillin, chloramphenicol, erythromycin, gentamicin, tetracycline, and vancomycin) against all tested strains [21]. Another paper has shown that *E. coli* eae+, *Pasteurella multocida*, and *S. enterica* species showed a higher sensitivity while the G+ *Streptococcus uberis* was the least sensitive. This paper also demonstrated the impact that size has on the antibacterial properties of the nanoparticles, by testing two different-sized Ag NPs. The MIC results found for Ag NPs 8 nm in size were half those of Ag NPs 28 nm in size for three of the six bacteria tested. The MICs found for these 8 nm nanoparticles were 12.5, 25, and 25 μg mL^−1^ for *S. enterica, S. aureus*, and *Actinobacillus pleuropneumoniae*, respectively. For the other three bacteria tested, *E. coli* showed a greater effect and a reduction of four times the MIC found at 28 nm (25 μg mL^−1^), *P. multocida* showed a MIC of 6.3 μg mL^−1^ with both nanoparticle sizes; however, *S. uberis* showed a higher MIC for smaller nanoparticles, 100 μg mL^−1^. The synergistic, additive, or indifferent activity between Ag NPs and routine antibiotics was also proved, however, these results vary according to the size of the nanoparticle, antibiotic studied, and strain used [22].

A greater bactericidal and bacteriostatic effect of biologically synthesized Ag NPs, concerning the solution of silver nitrate, the precursor of these nanoparticles, was found against multidrug-resistant strains of *Salmonella Typhimurium*, isolated from animals with diarrhea and septicemia. This highlights the potential application of Ag NPs as an alternative against multidrug-resistant strains [23]. At the concentration of 25 μg mL^−1^, Ag NPs completely inhibited the *Salmonella* strain, proving an effective antibacterial potential [24]. Mycosynthesized silver nanoparticles by *A. brasiliensis* exhibited an average particle size of 35.8 nm and these NPs showed a minimum inhibitory concentration of 300, 400, 500, and 600 μg mL^−1^ for *Bacillus subtilis*, *S. aureus*, *E. coli*, and *P. aeruginosa*, respectively [25].

Ag NPs, as well as ZnO NPs, at different concentrations of 5.31 μg mL^−1^ and 1.5 mg mL^−1^, respectively, showed activity against multidrug-resistant bacteria causing relevant veterinary pathologies, *Staphylococcus aureus*, *Salmonella enterica* subsp. Bukuru, and *Escherichia coli*. Another positive factor for the use of nanoparticles was the investigation of the synergistic effect between these and reference antibiotics [26]. Commercial ZnO NPs exhibit a dose-dependent inhibitory effect against the production of virulence factors, such as pyocyanin and *Pseudomonas* quinolone signal (PQS) production by *Pseudomonas aeruginosa*; the MIC was 300 mM. In addition, ZnO NPs inhibit human red blood cell hemolysis by *P. aeruginosa* [27]. The bactericidal action of Ag NPs and ZnO NPs was also detected against fish pathogenic microorganisms, such as the bacteria *Aeromonas hydrophila* and *A. salmonicida*; for *Yersinia ruckeri*, only the bactericidal action of ZnO NPs was observed [28]. Other important inorganic nanoparticles commonly studied are synthesized Cu NPs, which had significant in vitro antibacterial activity against *E. coli*, *K. pneumoniae*, *P. aeruginosa*, and *S. Typhi*, in comparison to the control [29].

### 2.2. Antifungal Activity

#### 2.2.1. Activity against Pathogenic Species

Mycosis is a term used to describe the fungal infection of animals, humans, and plants. These diseases are challenging due to the restricted number of therapeutics, the extended time of treatment, and the appearance of resistant isolates. Due to these circumstances and the antimicrobial potential of nanoparticles, several studies have been conducted to show the antifungal potential of different NPs.

Candidiasis is an opportunistic mycosis that can occur in different animals. This disease is caused by different *Candida* species. The antifungal activity of silver nanoparticles (4.1 ± 1.44 nm) was shown against the growth of *C. albicans*, *C. glabrata*, *C. krusei*, and *C. parapsilosis*, however, there is no significant effect against *C. tropicalis* [30]. Jalal et al. (2019) found similar results using biosynthesized Ag NPs against *C. albicans*, *C. dubliniensis*, *C. krusei*, *C. parapsilosis*, and *C. tropicalis*. However, the MIC ranged from 125–250 μg mL^−1^ and the minimal fungicidal concentration (MFC) was 500 μg mL^−1^; this concentration could be due to the average size of the nanoparticles that ranged from 10 to 100 nm. Ag NPs also interfere with some virulence factors by inhibiting the production of extracellular hydrolytic enzymes such as phospholipases, secreted aspartyl proteases (SAPs), lipases, and hemolysins, and by impeding germ tube formation [31]. This MIC was similar to the one found in another study, in which mycosynthesized Ag NPs (with an average size of 35.8 nm) had a MIC of 300 μg mL^−1^ against *C. albicans* growth [25].

Iturin (It) is a lipopeptide antibiotic manufactured by *Bacillus* spp. strains and this substance can efficiently control candidiasis infection in mice. Iturin-Ag NPs showed a MIC of 2.5 μg mL^−1^ against 10^6^ colony-forming units/mL (CFU/mL) of *Candida albicans*. The mechanisms of these inhibitions might be due to the increased permeability and loss of the integrity of the cell membrane and the induction of reactive oxygen species (ROS) in a time-dependent manner [32]. Copper oxide nanoparticles (CuO NPs) merged with polycaprolactone (PCL), a biopolymer used as a vehicle, exhibited antifungal activity against and induce changes in the morphological structure of *C. albicans*, *C. glabrata*, and *C. tropicalis*, initially at 25 mM in a dose-dependent manner [33].

Another pathogenic and important yeast, *Cryptococcus neoformans*, was susceptible to gold nanoparticles (Au NPs) and Ag NPs, with average sizes of 4.1 and 2.22 nm, respectively. This antifungal effect occurred in a time- and concentration-dependent manner, with a decrease in the percentage of viable *C. neoformans* cells [30]. Cryptococcosis is a systemic mycosis that can lead to cryptococcal meningoencephalitis, however, the established treatment, amphotericin B (AmB), might not reach significant concentrations in the central nervous system. Therefore, the conjugation of AmB with streptavidin merged with Au NPs (Au NPs-Sa-AmB) was used at the concentration of 0.04 μg mL^−1^ and effectively inhibited *C. neoformans* growth in vitro. In addition, this conjugated drug at a dosage average of 0.25 mg Kg^−1^ induced significantly reduced cryptococcal burden in the brain tissue of mice infected intravenously and treated with the same concentration of amphotericin B [34].

Dermatophytosis is a superficial mycosis common in animals and mainly caused by *Microsporum* spp., *Nannizzia* spp., and *Trichophyton* spp. Ag NPs with a size of 4.1 ± 1.44 nm at the concentration of 30 μg mL^−1^ induced significant growth inhibition of three important dermatophyte species, *Nannizzia gypsea* (prev. *Microsporum gypseum*), an important pathogen that usually leads to dermatophytosis in small animals, *Trichophyton mentagrophytes*, a species that causes disease in large animals, and *Trichophyton tonsurans* [30].

The oomycete *Pythium insidiosum* causes pythiosis, a serious disease that affects mammals, especially horses, in Brazil. Biogenic silver nanoparticles tested against *P. insidiosum* isolates showed a MIC50 of 0.24 μg mL^−1^ and a MIC90 of 0.47 μg mL^−1^. In addition, there were changes in the morphology, such as roughness of the hyphal surface area, retraction and loss of continuity of the cell walls, as well as the presence of disruption areas [35]. *Aphanomyces invadans* is another oomycete that can cause fish epizootic ulcerative syndrome. A strain of this species was isolated from a dwarf gourami fish (*Colisa lalia*) and tested against Ag NPs and ZnO NPs; the minimum inhibitory concentration obtained was 3.15 μg mL^−1^ and 17 μg mL^−1^, respectively [28].

#### 2.2.2. Activity against Mycotoxigenic Species and Mycotoxin Production

Mycotoxins are toxic secondary metabolites produced by different fungal species; the main genera are *Alternaria* spp., *Aspergillus* spp., *Fusarium* spp. and *Penicillium* spp. These fungal genera can produce the main relevant mycotoxins that can lead to several health problems and economic losses for animals [36]. Jain et al. (2020) have studied the use of ZnO NPs of sizes ranging from 15 to 20 nm against the growth of *A. alternate*, a potential producer of alternariol, alternariol methyl ether, and tenuazonic acid. The results showed a significant antifungal effect (83.93% mycelial inhibition) and a 92.22% reduction of spore germination at 250 μg mL^−1^ [37]. The antifungal effects of spherical ZnO NPs, with an average size of 40 nm, against *Aspergillus niger* and *Botrytis cinerea* were concentration dependent. At the lowest concentration tested, 6 mmol L^−1^, the reduction of fungal growth was higher than 60%, thus at 12 mmol L^−1^ more than 98% was inhibited completely for *A. niger* and *B. cinerea* [38].

Within the genus *Fusarium* spp. there are important species that are producers of fusariotoxins. Shende et al. (2015) studied three of the most relevant, *F. culmorum*, *F. oxysporum*, and *F. graminearum*, and discovered that these strains were sensitive to ~33 nm green synthesized Cu NPs impregnated with sterile discs [29]. The inhibitory effect of Ag NPs at different concentrations was evaluated against the mycelial growth of *F. graminearum* in potato dextrose agar (PDA) or potato dextrose broth (PDB). The antifungal effect was concentration dependent, with inhibitions of 80.56% and 86.78% at the highest concentration tested (20 μg mL^−1^) in PDA and PDB medium, respectively. The authors also observed a suppression of spore germination and germ tube growth. This paper also showed deoxynivalenol (DON) inhibition caused by the action of Ag NPs at all the concentrations tested [39].

Ag NPs ranging from 14 to 100 nm had a dose- and time-dependent antifungal effect against eight *Fusarium* species. The most susceptible strains showed an effective dosage to inhibit spore viability and colony growth of 2 μg mL^−1^ for *F. graminearum*, *F. langsethiae*, and *F. poae*, 10 μg mL^−1^ for *F. sporotrichioides*, and 15 μg mL^−1^ for *F. culmorum*, after 30 h of exposure. In addition, these Ag NPs also had a dose- and time-dependent antimycotoxin effect against deoxynivalenol (DON), 3–acetyl DON derivative (3–AcDON), and zearalenone (ZEA) produced by *F. graminearum* and *F. culmorum*; T-2 and HT-2 toxins produced by *F. sporotrichioides* and *F. langsethiae*; and fumonisins B_1_ and B_2_ produced by *F. proliferatum* and *F. verticillioides*. However, the production of nivalenol by *F. poae* only significantly decreased dependent on the increase in concentration [40]. Silver nanoparticles with a similar range in size from 20 to 100 nm displayed an antifungal effect against mycotoxigenic species affected by dose and/or exposure time. The spores of aflatoxigenic strains, *Aspergillus flavus* and *A. parasiticus*, were more resistant to Ag NPs than the ochratoxigenic (*A. carbonarius*, *A. niger*, *A. ochraceus*, *A. steynii*, *A. westerdijkiae*, and *Penicillium verrucosum*) species. An effective dose to inhibit spore germination and fungal growth of *A. flavus* and *A. parasiticus* was 30 μg mL^−1^ and 45 μg mL^−1^, respectively, for 20 h. However, it was 15 μg mL^−1^ for 20 h for ochratoxigenic strains. The production of aflatoxins B_1_, B_2_, G_1_, and G_2_ by aflatoxigenic strains was significantly influenced by the concentration and exposure time, as well as the production of ochratoxin A by the ochratoxigenic species [41]. This reveals that different species and even different strains can lead to divergent results, as well as other factors, such as exposure time, concentration, and characteristics of the nanoparticles.

Biogenic Ag NPs, nanoparticles synthesized using biological organisms (green synthesis), that are larger in size (average of 90 nm) than the other nanoparticles cited previously, were able to inhibit the growth of *Aspergillus* spp. This genus has great importance in the production of mycotoxins and is the main cause of an important fungal disease called aspergillosis, which is most common in poultry production. The MIC of *A. flavus*, *A. nomius*, and *A. parasiticus* was 8 μg mL^−1^; however, the MIC of *A. melleus* and *A. ochraceus* was lower, 4 μg mL^−1^. The MFC for *A. flavus* was 4-fold higher than the MIC, and for *A. nomius*, *A. parasiticus*, *A. melleus*, and *A. ochraceus* it was 8-fold greater. In addition, scanning electron microscopy revealed severe damage to hyphal morphology and conidial germination of *A. flavus* and *A. ochraceus* [42]. Khalil et al. (2019) also proved the antifungal effect of biogenic Ag NPs produced by *Fusarium* spp. (FAg NPs) and *Penicillium* spp. (PAg NPs) against *Aspergillus* spp. The MIC values of FAg NPs and PAg NPs were, respectively, 48 and 45 μg mL^−1^ for *A. flavus* and 51 and 47 μg mL^−1^ for *A. ochraceus*. This study also showed that Ag NPs could damage fungal membranes and lead to a leakage of proteins and DNA. In addition, the inhibition of AFs and OTA production by the *Aspergillus* species tested was statistically proven, ranging from 5.6 to 6.3 μg mL^−1^. This paper shows that the size of the nanoparticles can interfere with the antifungal capacity, with the smallest nanoparticles being more active than the bigger ones [43]. Ecofriendly synthesized copper oxide nanoparticles (CuO NPs), with an average size of 33 ± 2 nm, at concentrations of 500 and 1000 μg mL^−1^ had a significant antifungal (more than 80% inhibition) activity against mycotoxigenic species, *A. flavus* and *A. niger*, probably due to the cytoplasmatic damage and apoptosis caused by these nanoparticles [44]. An interesting point about these results is that the nanoparticles synthesized by biological methods show great variations of shapes and sizes and, in consequence, the activity.

Selenium nanoparticles (S NPs) at a concentration of 200 μg mL^−1^ had antifungal activity against *Alternaria* isolates, *Fusarium verticillioides*, and *Fusarium graminearum*, however, Trichoderma-derived selenium nanoparticles (TS NPs) showed higher activity against the growth rate and marginal hyphal deformation of the fungi tested. This was explained by the differences between TS NPs and SNPs; more than 30 metabolites were identified capping the selenium nanoparticles and more than 25 had a potential antifungal activity. In addition, these nanoparticles reduced fumonisin B_1_, tenuazonic acid, alternariol, and deoxynivalenol formation and altered synthetic gene expression [45].

### 2.3. Antibiofilm Effects

Biofilm is defined as an aggregate of microorganisms accumulated on solid–liquid surfaces, biological or not, coated with a matrix of polymeric extracellular substances [46]. Biofilm formation is a virulence mechanism of pathogenic species, and this has a relevant impact in the veterinary field, especially animal production. Therefore, the ability to inhibit or reduce biofilm formation and/or activity is a relevant feature, which is observed in different substances and particles.

Antibiofilm formation can be identified while using silver nanoparticles, with a minimum biofilm inhibitory concentration (MIBIC50) of 64 μg mL^−1^ against *Streptococcus agalactiae*; interestingly, the combination of Ag NPs and cinnamon oil improved the antibiofilm activity against some strains tested [8]. Ag NPs also have antibiofilm activity against *Prevotella melaninogenica* and *Arcanobacterium pyogenes*; the biofilm formation decreased by more than 95% and 90%, respectively. This study also indicates that the induction of oxidative stress and the disruption of cell membranes could be the mechanism of action of Ag NPs [20]. This same Ag NP antibiofilm activity was also detected regarding the growth of *S. agalactiae*, *S. dysagalactiae*, *Salmonella* spp., *E. faecalis*, *E. cloacae*, *E. coli*, and *S. aureus* and the yeast *Candida albicans* at a concentration of 200 μg mL^−1^. The same Gram-positive and Gram-negative pathogenic bacteria also showed the same susceptibility under the same concentration of Ag-Cu NP complexes [4]. Ag NPs derived from *E. coli* (EcAg NPs) and *Curcuma longa* (ClAg NPs) showed a significant antibiofilm effect against *P. aeruginosa* but not for *S. pseudintermedius* [11].

Silver nanoparticles decorated with quercetin nanoparticles (QA NPs) highly inhibit biofilm formation by *E. coli* ECDCM1 strains at 20 μg mL^−1^ on glass, possibly by reducing the transcription of biofilm-associated genes [12].

Chitosan nanoparticles, which are biological NPs, also showed an inhibitory effect on biofilm formation, in a size- and dose-dependent manner for strains of *Staphylococcus* sp.; the relationship between size and inhibition was more evident as the concentration increased [19]. Chitosan nanoparticles (Qo NPs) act as antibiofilm producers of *Pseudomonas* spp., inhibiting more than 70% of biofilm formation at a concentration of 70 μg mL^−1^ at room temperature, 25 °C. However, due to the thinner and less dense biofilm at 37 °C, the percentage of inhibition increased up to 90% [18].

Commercial ZnO nanoparticles in dose-dependent treatment can inhibit the biofilm formation and cell growth of *E. coli* O157:H7 and *S. aureus*. At 1 mM, these ZnO NPs almost completely inhibited the *P. aeruginosa* biofilm formation on a glass surface [27]. ZnO NPs also showed desirable inhibitory and bactericidal effects against multidrug-resistant *S. aureus* and *E. coli* bacteria isolated from the milk of ewes affected by mastitis; in addition, the study allowed us to verify a greater effect of ZnO NPs on Gram-positive (G+) than Gram-negative (G−) bacteria [14].

Inorganic, organic, polymeric, lipid, nanotube, and other nanoparticles are being studied as novel approaches for wound healing and regenerative medicine to improve wound healing in different tissues. One of their capacities is antibiofilm activity, that can help reduce or even inhibit biofilm formation, which is effective in helping overcome microbial contamination. The mode of action and specific characteristics are directly dependent on the nanoparticle used and it is very important to note that the use of an innocuous nanoparticle is essential for this application [47,48].

### 2.4. Antiviral Effects

Rift Valley fever virus (RVFV) is a viral pathogen that causes disease in ruminants and is transmitted to humans during epizootic outbreaks. Currently, there is no available treatment or vaccine; therefore, new therapeutic methods such as nanoparticles are constantly being studied. Commercial Ag NPs reduced approximately 50% of the final virus production compared to control; they also reduced the viral infectivity in a dose-dependent manner. The greatest reduction, 98%, was reached with preincubation of RVFV at 12 μg mL^−1^. In vivo tests were also conducted and the group of mice that received a pretreatment had a survival rate statistically higher than the control group [49].

African swine fever virus (ASFV) is a highly contagious and fatal virus and there is no vaccine or effective medicine available. However, silver nanoparticles at 0.78 μg mL^−1^ completely inhibited this virus at a titer ≤ 10^3^ with 50% hemadsorbing doses (HAD50) [24]. Another important virus with a lack of prevention or lifelong treatment is herpes simplex virus (HSV), an endemic virus that can infect different animal species and humans. The antiviral effects of garlic acid-stabilized highly mono-dispersed gold nanoparticles (GAu NPs) were tested against HSV types 1 and 2. As a result, a dose-dependent reduction of viral plaque formation and interference in the infection proliferation were detectable [50].

Influenza type A virus (IAV) can affect different animal species, such as cats, horses, pigs, chickens, ducks, whales, and seals. Porous gold nanoparticles (PoG NPs), after 60 min of exposure at 0.2 mg mL^−1^, were able to successfully inactivate IVA H1N1 and H3N2 and reduce viral infection of IVA H9N2 [51]. As well as the previously cited study, the postexposure of H1N1 with 75 μg mL^−1^ of ZnO NPs and polyethylene glycol-coated ZnO NPs (ZnO-PEG-NPs) showed a reduction of 2.2 log_10_ and 1.2 log_10_, respectively. The same study also showed a dose-dependent antiviral activity [52]. The transcript levels of H1N1 viral RNA decreased ~100 fold compared to control in the presence of 10 μM of iron oxide nanoparticles (Fe_3_O_4_ NPs) at 24 h postinfection [53].

### 2.5. Antiparasitic Effects

Parasite infestation is a major problem in livestock production, due to the direct and indirect productivity losses and the treatment and control costs. Antiparasitic resistance has already been identified in many different species of ticks, helminths, and other parasites [54]. Even though there are well-established guidelines to manage resistance in parasite control in ruminants, resistance is still a problem in animal production [55]. Due to the resistance, a worldwide phenomenon, the study of nanoparticles isolated and combined with other drugs is being conducted as an alternative method to control ticks. One of these studies showed the advantages of coating silver nanoparticles with deltamethrin drug and neem essential oil extract (DN-Ag NPs); the results showed a significant decrease in the lethal concentration needed to eliminate 50% (LC50) and 99% (LC99) of the larvae and adults of *Rhipicephalus* (*Boophilus*) *microplus*. In addition, this study also showed that DN-Ag NPs could interfere with the reproductive performance of adult female ticks, by acting as an ovipositor inhibitory compound [56].

Another paper showed the antiparasitic potential of nickel nanoparticles (Ni NPs) against important vectors of diseases. The LC50 values of Ni NPs against larvae of *R.* (*B.*) *microplus* and *Hyalomma anatolicum* were 10.17 mg L^−1^ and 10.81 mg L^−1^, respectively. The larvicidal activity of this nanoparticle was also proved against the fourth instar larvae of *Anopheles subpictus*, *Culex quinquefasciatus*, and *C. gelidus*, with the LC50 being 4.93, 5.56, and 4.94 mg L^−1^, respectively [57]. Green synthesized silver nanoparticles (Ag NPs) also showed a lethal effect against larvae of *R.* (*B.*) *microplus* with an LC50 of 8.98 mg L^−1^, but also a larvicidal potential against larvae of *A. subpictus* and *C. quinquefasciatus*; the LC50 found was 13.90 and 11.73 mg L^−1^, respectively [58]. ZnO NPs have also been tested against the larvae of the same parasites; the LC50 was 13.41 mg L^−1^, 3.19 mg L^−1^, and 4.87 mg L^−1^ for *R.* (*B.*) *microplus*, *A. subpictus*, and *C. quinquefasciatus*, respectively [59].

### 2.6. Drug Delivery Systems

#### 2.6.1. Antimicrobial and Antiparasite Delivery Systems

Poly (lactide-co-glycolide) (PLGA) is a biocompatible nanoparticle-encapsulated delivery system commonly used as a drug delivery system. The combination of PLGA and other molecules can improve the capacity of these substances to act as a carrier. *Poly*(ethylene glycol)-block-poly(lactide-co-glycolide) (PEG-PLGA) is a nanocompound used for drug encapsulation and delivery systems. A study has used this substance as a nanocarrier of diphyllin, a vacuolar ATPase inhibitor. The diphyllin-loaded nanoparticles presented a broadened therapeutic window, showing a 13-fold reduction of cytotoxic concentration and 60-fold higher antiviral activity compared to free diphyllin. In addition, these nanoparticles reduced, in a dose-dependent manner, feline infectious peritonitis virus (FIPV) infection in vitro; the in vivo study also showed toleration of the tested mice upon high-dose intravenous administration [60]. Another study showed an in vitro dose-dependent antiviral and anti-inflammatory effect of curcumin-encapsulated chitosan nanoparticles (Cur-CS NPs) against FIPV infection; at the concentration of 20 μM, this compound showed a significant protective effect on virus-infected cells at pretreatment, cotreatment, and posttreatment. In addition, this formulation improves oral absorption in healthy cats [61].

Nanospheres and nanocapsules of 2-amino-thiophene (6CN10) and 6CN10 complexed with 2-hydroxypropyl-β-cyclodextrin (6CN10:HP-β-CD complex) were tested against yeasts, but only *Cryptococcus* strains were susceptible to these nanoparticles. The nanocapsules of 6CN10 and 6CN10:HP-β-CD complex exhibited MIC values of 83.33 μg/mL and 0.1–0.2 μg/mL, respectively. However, the best results were obtained from the nanospheres, as 6CN10 had MIC values of 41.66 to 0.32 μg/mL, and 6CN10:HP-β-CD complex showed MIC values of 0.2 to 0.1 mg/mL [62].

The encapsulation of miltefosine (MFS), a synthetic drug used to treat leishmaniasis and other diseases, in alginate nanoparticles (Alg NPs) can make the release of the drug sustainable, by slowly and constantly releasing it. Despite the higher in vitro MIC values of the miltefosine-loaded Alg NPs (MFS-Alg NPs) against *Candida* spp. and *Cryptococcus* spp. strains, these nanocarriers are attractive due to the lower toxicity found [63]. Solid lipid nanoparticles (SLNs), nanocarriers formed by a solid lipid matrix, as well as the MFS-Alg NPs, show a sustained release. Entrapped terbinafine hydrochloride in SLNs is released over more than 24 h. In addition, SLNs enhance drug retention, reducing systemic drug diffusion and side effects, which elucidates the advantages of using this carrier for topical administration of drugs [64]. Another study has used SLNs as a clotrimazole (CLZ) carrier and as a dual-drug delivery carrier with clotrimazole and alpha-lipoic acid (ALA), a substance that could improve the effectiveness of CLZ. The study has also tested the behavior of positively charge-loaded SLNs, and the results proved that despite the controlled release diffusion in vitro, CLZ-SLNs and ALA-CLZ-SLNs had the same antifungal activity as free clotrimazole molecules against *Candida albicans* strains. However, the cationic SLNs loaded with CLZ had an improved activity [65].

Liposomal nanoparticles (LP NPs) are a promising drug delivery system, as the encapsulation of these carriers with antifungal drugs such as anidulafungin, a less toxic antifungal than echinocandin, increases drug activity, reduces toxicity, and improves pharmacokinetics. The minimum inhibitory concentration of LP NPs loaded with anidulafungin is 12.50, 6.25, and 1.56 µg mL^−1^, respectively, for 0.32 ± 0.01% weight/weight (wt/wt), 1.59 ± 0.05% (wt/wt), and 2.83 ± 0.53% (wt/wt) anidulafungin liposomal formulations [66].

Ellagic acid (EA) is a phenolic compound that has interesting antioxidant characteristics and antimalarial properties; however, this molecule is unstable and has poor bioavailability and solubility. Due to these characteristics, Beshbishy et al. (2019) identified the in vitro dose-dependent growth inhibitory potential of EA-loaded nanoparticles (EA-NPs) against the growth of *Babesia bovis*, *B. bigemina*, *B. divergens*, *B. caballi*, and *Theileria equi*. In addition, EA-NPs showed additive and synergistic effects when administered with diminazene aceturate (DA) or atovaquone (AQ). In vivo experiments in mice showed a significantly reduced parasitemia peak of *B. microti* when treated with 70 mg mL^−1^ of EA-NPs [67].

Nanoscale polymers are widely used in drug delivery systems. Zein is a natural polymer that is biodegradable, resistant, and low cost. The encapsulation of eugenol and garlic essential oil, botanical compounds with antibiotic characteristics, in zein nanoparticles has increased the antibiotic activity against important fish pathogens, *Aeromonas hydrophila*, *Edwardsiella tarda*, and *Streptococcus iniae*. Another important discovery is that the nanoencapsulation reduced the toxicity of these botanical compounds by increasing the LC50 [68].

#### 2.6.2. Vaccine Delivery System

Besides the use of PLGA nanoparticles (PLGA NPs) as an antimicrobial delivery system, these nanoparticles are frequently used as an adjuvant of vaccines, because they can enhance the activity and improve the immune response. Avian influenza virus (AIV) is a pathogenic virus that causes either mild clinical symptoms or massive outbreaks with high mortality. The available vaccines usually do not prevent virus circulation and infection. The encapsulation of inactivated AIV with PLGA NPs (NanoAI) and the co-encapsulation of inactivated AIV and CpG-ODN, a vaccine adjuvant that increases efficacy, with PLGA-NPs (NanoAI + CpG) were studied. Subcutaneous vaccination revealed that co-encapsulation of CpG-ODN improved vaccine formulation; also, higher hemagglutination inhibition titers (HI) were observed while using prime vaccination of NanoAI + CpG. In addition, the mucosal application of mannan and chitosan-coated NanoAI + CpG induces a good antibody response [69].

Killed porcine reproductive and respiratory virus was entrapped in PLGA and the results demonstrated a potential to elicit an immune response. The virus-neutralizing titers in lung homogenate of vaccinated pigs were significantly higher than controls; in addition, there was an increase in the frequency of total lymphocyte population, IgA levels, and Th1 cytokines (IL-12 and INF-γ) and a decrease in IL-4, IL-6, IL-10, Th2, and TGF-β proinflammatory cytokines [70]. Dhakal et al. (2018) have also used chitosan as an adjuvant in vaccines. A vaccine composed of a nanoparticulate mucoadhesive polymer of chitosan (CNPs) encapsulated with swine influenza A virus (CNPs-KAg) was tested in vitro and in vivo. Porcine monocyte-derived dendritic cells treated with CNPs-KAg showed a higher production of cytokines than control in vitro. In addition, the in vivo study showed enhanced induction of secretion of antibodies and cross-reactive lymphocytes by CNPs-KAg compared to control, which resulted in a reduction of nasal virus shedding, pulmonary viral titers, and inflammatory changes in lungs [71].

Bovine respiratory syncytial virus (BRSV) leads to high morbidity in calves. Nanovaccines based on mucosal polyahydride co-encapsulated with proteins showed an interesting induction of inflammatory cytokines in vitro. In vivo, this single vaccination could induce specific antibodies in the respiratory tract, reducing gross lung lesions and the area affected. Additionally, histological lesions were diminished compared to the control group as well as quantities of viral burden and shedding [72].

Reports have also described the use of biological nanoparticles as adjuvants in nanovaccines. Protein nanoparticles (SANPs), which are composed of protein monomers and can mimic structures of small viruses, were also tested against infectious bronchitis virus (IBV) [73]. Another paper evaluated the potential of a self-assembling recombinant rotavirus VP6-ferritin nanoparticle vaccine produced in mammal glands and concluded that there was an induction of the protective effect and immunogenicity [74].

The vaccination of animals against parasites is important to control some diseases, such as *Echinococcus granulosus*. An in vitro experiment proved that canine monocyte-derived dendritic cells (cMoDCs) challenged with polymeric nanoparticles encapsulated with recombinant *Echinococcus granulosus* antigen (tropomyosin EgTrp) and adjuvanted with monophosphoryl lipid A (MPLA) showed an improvement in T cell proliferation [75]. Another paper also showed an immunogenic effect of nanostructures to improve the capacity of the sporozoite p67C antigen of *Theileria parva* [76].

## 3. Conclusions

As shown in this review, the antimicrobial properties of NPs are constantly being studied, however, each study uses different nanoparticles. The NPs can be synthesized in different manners and using innumerous compounds, and these differences give rise to NPs with contrasting characteristics, such as size, shape, affinity, oxidative and electric properties, agglomeration capacity, and others; these distinct attributes can lead to specific modes of action and variable antimicrobial potentials. This can be an obstacle to the study and development of future antimicrobial products since there is no standard for any of the nanoparticles synthesized and used in research. Nanoparticle studies have demonstrated a promising future for the use of these nanostructures as new drugs, vaccines, and drug delivery systems and even as an alternative method to maintain food quality and safety by reducing food contamination. The NP studies in the veterinary field are relatively recent, and more studies need to be conducted to elucidate the mechanisms of action and the future impact of these nanostructures on animal, human, and environmental health in the short and long term. However, the importance and the promising applications of these nanoparticles in the future of veterinary pharmacology are undeniable.

## Figures and Tables

**Figure 1 antibiotics-12-00958-f001:**
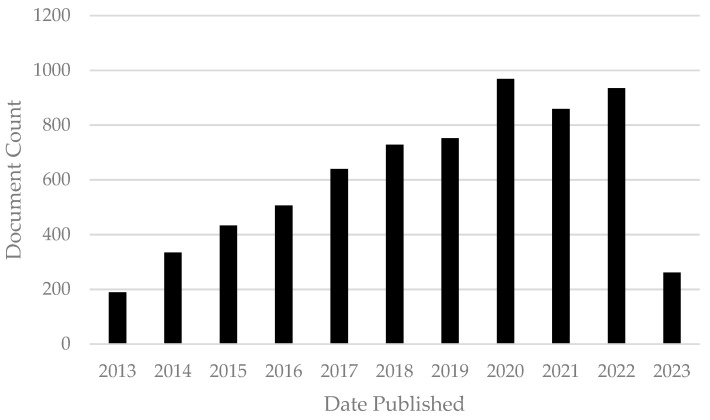
Document count of scholarly works from 2013 to 2023 obtained using the key words “nanoparticles AND veterinary” in the Lens.org database (accessed on 17 May 2023).

**Figure 2 antibiotics-12-00958-f002:**
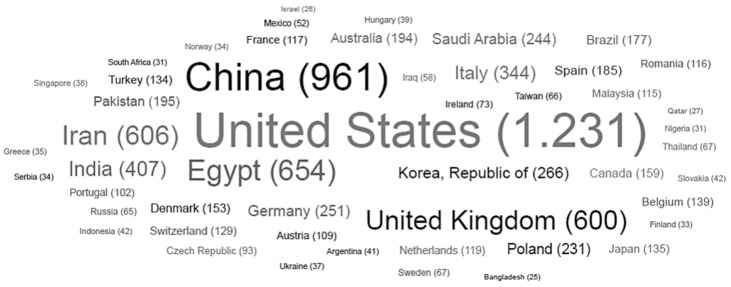
Word cloud showing the 50 countries that have the greatest number of publications from 2013 to 2023 obtained using the search term “nanoparticles AND veterinary” in the Lens.org database (accessed on 17 May 2023).

## Data Availability

Not applicable.

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
