# Peer review of "The Antimicrobial Applications of Nanoparticles in Veterinary Medicine: A Comprehensive Review"

_antibiotics, 2023, doi:10.3390/antibiotics12060958_

Round 1
Reviewer 1 Report
Reviewer comments
Rodrigues et al reviewed the antimicrobial applications of nanoparticles in Veterinary medicine. The author discussed clearly about the antibacterial activity, antifungal activity, antibiofilm effects, antiviral effects, antiparasitic effects, drug delivery systems. Overall the manuscript is very well written but requires some minor changes before Publication.
1. Keywords should be given in alphabetical order
2. What is the novelty of the work?
3. What is the importance of this work?
4. I suggest the author to provide methodology for the synthesis of nanoparticles in separate section
5. I suggest the author to add some points related to the diseases in animals caused by microbes
6. Author should explain more about Veterinary medicine
7. The typos should be corrected
8. I suggest the author should remove the highlight in section 2.2.2
9. I suggest the author to include some more points in introduction section
10. There is lack of information in the conclusion section. Author should include some more information in the conclusion part.
11. I suggest the author to add table which is relevant to the work
12. I suggest the author to insert the schematic representation related to the methodology and application
The minor editing of English language is required
Reviewer 2 Report
It is a very well written article. The topic is very novel. However, here are few of my comments.
1) I suggest the authors to merge conclusion and future directions section instead of splitting. Also, please revise the article carefully to resolve grammatical errors.
2) At the end of introduction, please remove the "Systematic review", since this is a comprehensive review.
3) In introduction, please include literature search methods along with number of publications available each year (for a period of last 10 years). Plot a bar graph. This will help readers witness the research growth in this area.
4) Most importantly, reproduce few in vitro and in vivo study results figures from previously published research articles. (At least 3 to 4 multicomponent figures).
5) Please discuss about the role of NPs in wound biofilm eradication (The biofilm in wounded area is generally formed due to bacteria, which slows down the wound healing process, thereby increasing the wound healing duration). Some suggestions to cite: Kushwaha et al. (2022) 10.3390/nano12040618, Gowda et al. (2023) https://doi.org/10.1016/j.mtchem.2022.101319
Minor English language editing is required.
Reviewer 3 Report
Dear authors, your manuscript entitled "The antimicrobial applications of nanoparticles in Veterinary Medicine: a comprehensive review" makes an extensive and timely overview of the situation to date regarding nanoparticles and their application.
Each small chapter on antibacterial, antifungal, antiparasitic, antibiofilm activity has been clearly inserted, with references to veterinary medicine as the purpose of the review.
The critical spirit of the authors is well evidenced by having stressed the critical points in the use, for now generally in vitro, of these NPs have also been underlined.
For the paragraph related to antibacterial activity, however, it would be interesting to provide some indication on the activity reported by the literature of the synergistic activity of NPs with antimicrobials (for example, Meroni G. et al., 2019).
Round 2
Reviewer 2 Report
The authors have addressed all the comments. The current form of manuscript can be accepted for publication in Antibiotics.
Minor English language editing is required.